# On a Vector towards a Novel Hearing Aid Feature: What Can We Learn from Modern Family, Voice Classification and Deep Learning Algorithms

**William Hodgetts [1,2,3], Qi Song [4], Xinyue Xiang [4] and Jacqueline Cummine [1,3,***

[1] Department of Communication Sciences and Disorders, University of Alberta, Edmonton, AB T6A2G4, Canada; william.hodgetts@ualberta.ca
[2] Institute for Reconstructive Sciences in Medicine, Covenant Health, Edmonton, AB T5R4H5, Canada
[3] Neuroscience and Mental Health Institute, University of Alberta, Edmonton, AB T6A2G4, Canada
[4] Department of Computing Sciences, University of Alberta, Edmonton, AB T6A2G4, Canada; qsong@ualberta.ca (Q.S.); xxiang2@ualberta.ca (X.X.)
* Correspondence: jacqueline.cummine@ualberta.ca; Tel.: +1-780-492-3965

**Abstract:** (1) Background: The application of machine learning techniques in the speech recognition literature has become a large field of study. Here, we aim to (1) expand the available evidence for the use of machine learning techniques for voice classification and (2) discuss the implications of such approaches towards the development of novel hearing aid features (i.e., voice familiarity detection). To do this, we built and tested a Convolutional Neural Network (CNN) Model for the identification and classification of a series of voices, namely the 10 cast members of the popular television show "Modern Family". (2) Methods: Representative voice samples were selected from Season 1 of Modern Family ($N = 300$; 30 samples for each of the classes of the classification in this model, namely Phil, Claire, Hailey, Alex, Luke, Gloria, Jay, Manny, Mitch, Cameron). The audio samples were then cleaned and normalized. Feature extraction was then implemented and used as the input to train a basic CNN model and an advanced CNN model. (3) Results: Accuracy of voice classification for the basic model was 89%. Accuracy of the voice classification for the advanced model was 99%. (4) Conclusions: Greater familiarity with a voice is known to be beneficial for speech recognition. If a hearing aid can eventually be programmed to recognize voices that are familiar or not, perhaps it can also apply familiar voice features to improve hearing performance. Here we discuss how such machine learning, when applied to voice recognition, is a potential technological solution in the coming years.

**Keywords:** machine learning; voice classification; hearing aid; voice familiarity

## 1. Introduction

There are many hearing aid features and advances that have been shown to improve outcomes for individuals with hearing loss. For example, they increase speech perception and comfort in background noise while also decreasing the effort required to listen [1]. The features include, but are not limited to, directional microphones [2–10] noise reduction algorithms [11], dynamic range compression [12] and proper fitting and verification [13–16]. What is notable is that each of these factors was conceptualized, tested, validated and incorporated into current hearing aid devices via digital signal processing (DSP) techniques, advanced hearing aid technology, and more recently, machine learning approaches [17–20]. In each case, the incorporated features are associated with increased identification, assessment and/or hearing performance across a variety of listening environments [17,19–22]. Here we propose a new potential feature, voice identification and conversion, to explore voice familiarity as a possible hearing aid feature as voice familiarity has been shown to have specific neural markers [23,24] and improve speech recognition in adults with [25,26] and without [27] hearing loss. However,

to our knowledge, the conceptualization of voice familiarity as a hearing aid feature has yet to be explored. Our motivation is as follows: if a hearing aid could be programmed to store a few (or even one) voice profile that is a highly familiar voice (e.g., spouse) the hearing aid would have stored the key features of that voice that make it familiar to the user. If, in the future, real time processing could be leveraged to enhance the features of a novel voice by applying the stored features of a familiar voice, it may be possible to leverage a non-hearing aid feature (voice familiarity) by taking advantage of the tiny computer(s) the patient has in their ears. Here, we aim to (1) expand the available evidence for the use of machine learning techniques for voice classification and (2) discuss the implications of such approaches towards the development of novel hearing aid features (i.e., voice familiarity detection).

### 1.1. Voice Familiarity: Improved Outcomes and Reduced Listening Effort

As we explore the potential for new hearing aid features, there are several steps and questions that need to be considered. For example, (1) conceptualization of potential features, (2) testing behavioral/cognitive outcomes associated with and without the feature, (3) proof of concept of the feature in technological space, (4) development of hardware and software to support the feature, and (5) validation and verification in a real-world setting. While this is not an exhaustive list of the necessary steps, and by no means happens in a purely linear fashion, it does provide a general guideline from which we can test claims about potentially useful hearing aid advancements. With respect to voice familiarity as a hearing aid feature, steps 1 and 2 have been well established [25,27,28]. For example, the age-old story of an individual with hearing loss who claims they have no trouble hearing their spouse across the table in a restaurant, but they do have trouble understanding the friendly patron at the table next to them, is one genesis of the notion that voice familiarity is an important feature to decreased listening effort. In recent work, this claim has been tested in individuals with and without hearing loss. Indeed, increased voice familiarity is associated with improved performance on complex dual-tasks, speech-in-noise tasks [25], and auditory recall [26,27]. For example, Johnsrude et al. [25] brought married couples into the lab to make recordings of signals to be used in the experiment with their partner. The researchers then played either their partner's voice as the target or another person's voice as the target in differing signal-to-noise ratios (SNRs). Predictably, performance improved as the SNR from the study increased, but, more importantly, performance when listening to their familiar partner was on average about 7 dB better than listening to an unfamiliar voice. Indeed, the cognitive and behavioral outcomes of a "familiar voice" compared to an "unfamiliar voice" are relatively stable in that there is enhanced performance [5,12,25–28].

### 1.2. Related Works

There has been much interest in the development of hardware and software to support various hearing aid features. Indeed, a simple search of "speech recognition and machine learning" returns >1 million articles, underscoring the substantial interest in this topic over the last five decades. Relevant to the current work, recently explored the application of machine learning in speaker identification to demonstrate feasibility of including additional voice features (i.e., dialects, accents, etc.) to enhance the security of voice recognition software (e.g., banking purposes). After the creation of a substantial database of voices, they reported a range of accuracy (81–88%) in speaker identification. Zhang et al. [29] explored a combined CNN + connectionist temporal classification (CTC) model to complete a phoneme recognition task and provided evidence that CNN models are an ideal approach for auditory based recognition applications, with accuracy at approximately 82%. While this paper was important for advancing our understanding of the feature extraction process (i.e., MFCCs), we focus here on providing additional support for deep neural networks, specifically the application of CNN, to speaker identification. While each of these studies were instrumental in advancing the machine learning literature, much more work is needed to determine consistency of findings (i.e., accuracy of training and testing) and to test the boundaries of the application of CNN to speaker identification (from/a/to words, to

multiple words, etc.). Nassif et al [30] published a systematic review to culminate the information with respect to applications of machine learning in the speech recognition domain. The authors observed that while 79% of the currently published papers tested automatic speech recognition, just 3% examined speaker identification.

### 1.3. Summary

Here we implement a machine learning approach to speaker identification to expand the available evidence in this space. In addition, we discuss the implications of such work in the context of eventually including "voice familiarity" as a feature in a hearing aid. In doing so, we provide additional support regarding the feasibility and consistency of a machine learning classifier to detect and recognize auditory input, in this case, the voices from the popular television series Modern Family. Furthermore, we provide a novel line of inquiry to be pursued by researchers in the hearing aid development space, namely capitalizing on the benefits of voice familiarity.

## 2. Materials and Methods

To build a machine learning model to accurately classify voices of varying characteristics we followed the procedure outlined in Figure 1: (1) Select representative voice samples, (2) Data cleaning of audio samples, (3) Feature extraction of audio samples, (4) Training of voice classification model, (5) Assessment of the accuracy of voice classification models (simple vs. advanced).

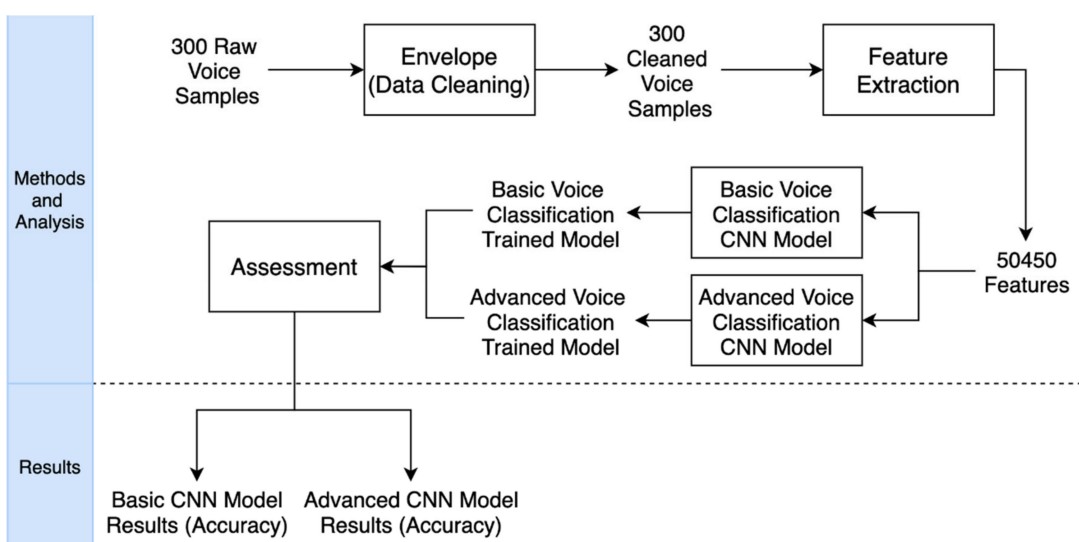

**Figure 1.** Pipeline for preprocessing and development of voice classification model.

### 2.1. Selection of Representative Voice Samples

We used "QuickTime Player" (Apple, Cupertino, CA, USA) software inside a MacOS Operating System to record audio samples and Switch software (NCH Software, Greenwood Village, CO, USA) to transfer the audio format to waveform audio file format. Specifically, we recorded 30 audio files from each of the 10 main actors in Modern Family Season 1, namely, Alex, Phil, Mitchell, Manny, Luke, Jay, Haley, Gloria, Claire, and Cameron, for a total of 300 voice recordings. The accumulated total length (i.e., the sum of the 30 audio recordings) for each individual can be found in Table 1. Only samples that contained a single voice/person talking were selected. The rationale for choosing these voices is that they exhibit diversity in gender, age, accent, are easily accessible, and provide future opportunities to explore developmental factors (i.e., there are 10 years/seasons worth of voices accessible across the age range).

**Table 1.** The total length of audio recordings sampled for each character in the Modern Family series.

| Name | Time (s) |
|---|---|
| Alex | 95.06 |
| Cameron | 103.28 |
| Claire | 100.93 |
| Gloria | 101.84 |
| Haley | 97.04 |
| Jay | 97.54 |
| Luke | 99.17 |
| Manny | 100.74 |
| Mitchell | 105.04 |
| Phil | 107.07 |

### 2.2. Data Cleaning of Audio Samples: Envelope

For data cleaning of audio samples, the envelop was adapted from Adams [31]. To maximize the quality of the training data, we removed extraneous noise initially inherent in the measurement of the raw audio file; see Figure 2A for a representative raw audio file. First, a "Librosa" function was used to read the audio data, and automatically resize the audio magnitude value from 0 to 1. An initial threshold was applied to remove artifacts from the audio file, namely tapered amplitudes both at the front end and back end of the audio samples, as well as significantly high amplitudes, likely resulting from an extraneous noise (i.e., telephone ringing). A mask with two thresholds, namely 0.001 and 0.999, corresponding to the minimum and maximum thresholds, was applied to the audio waveform, and the remaining voice envelope extracted (Figure 2B). We applied these thresholds via an implemented Envelope Function (see Appendix A) to remove audio under/over the specified noise threshold. The Envelope has two dimensions, the horizontal axis is time, and the vertical axis is amplitude.

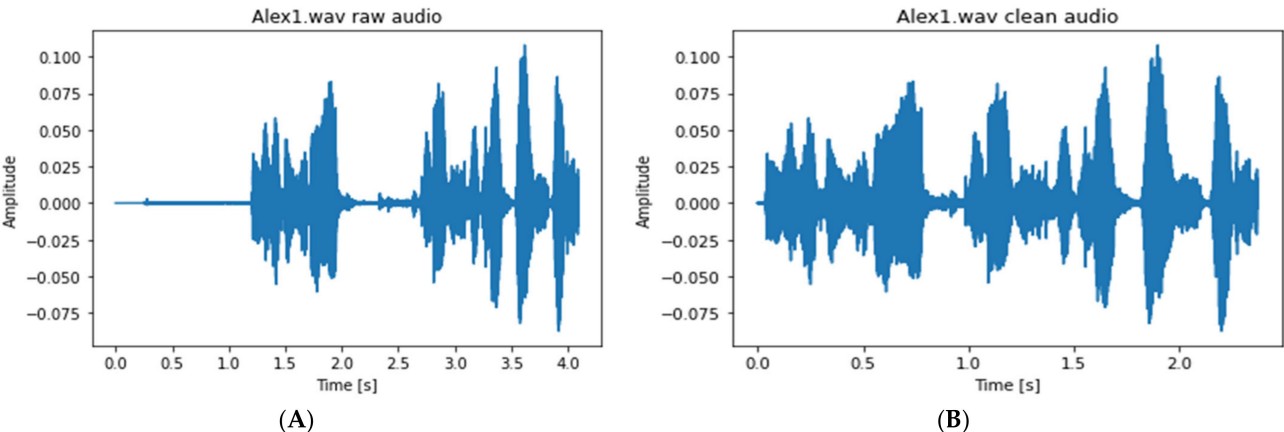

**(A)** **(B)**

**Figure 2.** (**A**) A representative raw audio file from the speaker, Alex, before thresholding. (**B**) A representative clean audio file from the speaker, Alex, after thresholding.

### 2.3. Feature Extraction

There are two main DNN architectures: convolutional neural network (CNN) and recurrent neural network (RNN) (Elman, 1990). We determined that the CNN approach was more appropriate for the current dataset as we had a limited number of samples from which we aimed to test our model. Previous work by You, Liu and Chen [32] observed that more complicated neural networks (i.e., RNNs and/or hybrid models) may result in lower accuracy and/or fail to converge when trying to model "relatively" small datasets and Zhang et al. [29] observed that RNNs are more computationally expensive compared to CNNs, with potentially little increases in accuracy [33]. Therefore we implemented a

convolutional neural network (CNN) model for voice samples machine learning training, which relied on image feature transformation (see [34] for a similar example). For each cleaned audio file, the spectrogram was extracted using the Mel-Frequency Cepstral Coefficient [35] algorithm, and subsequently formed the image feature used in the CNN model (see [32] for an example of this approach applied to audio data). We had a total of 1009 seconds of audio from the 300 audio files (i.e., 10 voices $\times$ 30 files). We multiplied 1009 by 50 (the number of epochs to be run) to get 50,450 samples for this experiment (to be divided into the training set, validation set and test set, discussed below). The following steps were used for feature extraction. First, we used the random function in the Python library to pick a random audio file from 300 audio files. Second, we used the audio processor function "scipy" library by McFee et al. [36] to read the audio file and return audio data and sample rate (in seconds). Third, we randomly sampled a section of the audio sample with a fixed window size inside the audio file. Fourth, we implemented an open-source code library, namely, "librosa.feature.mfcc" API McFee et al. [36]; source code can be found through https://librosa.org/doc/main/generated/librosa.feature.mfcc.html, accessed on 2 June 2021) to extract the spectral features into a matrix. The MFCC method extracts cepstrum from the audio data and the format of cepstrum we chose was 32 pixels by 32 pixels. We set the parameters as follows: sampling rate = 44,100, hop length = 700 n_mfcc = 32, n_fft = 512 (Appendix A). Therefore, the shape of matrix X is (50,450,32,32, 1) with each of the 50,450 cepstrum representing one feature of one person. Third, the CNN model requires the addition of one color channel (gray) to our matrices, as CNN models always train image data, and for images, there are three color channels for one dimension (RGB). We used one-hot encoding [37] to deal with the classes because our classes were not pure numbers but were instead people's names. One-hot encoding aids the machine model to train and predict each class as 0 or 1. For example, if the voice is from Alex, and the index of Alex is 5 (start from 0), the one-hot encoding matrix is [0,0,0,0,0,1,0,0,0,0]. Finally, we separate the 50,450 samples, with 80% of them being used as training data, 10% as validation data, and the remaining 10% as testing data as per TensorFlow documentation (available online https://www.tensorflow.org/tutorials/audio/simple_audio, accessed on 2 June 2020). Because we had 10 classes in this model, each class had about 4000 training samples, 500 validation samples, and 500 testing samples.

Convolutional Neural Network Model: Training

According to Cornelisse [38], to train the CNN model we first input the samples from the MFCC, as described above. In the feature learning stage, the CNN model learns the features which are extracted from the audio data via the function extract feature (Figure 3). In this feature learning period, there are several convolutions with activation function "ReLU". The basic idea of convolution here is multiplication and adding the three dimensions, height (in pixel), width (in pixel), and gray channel, respectively. Therefore, we pick each $1 \times 1$ pixel as a single piece, and every single piece has a gray color value from 0 to 255. In addition, there are $3 \times 3$ pixel filters that also have values in each pixel. The filter slides over the input and performs its output on the new layer [38]. Then matrix multiplication and addition are applied from the left-top corner of the cepstrum down to the right-bottom corner, then it will produce a destination image with those new pixels. This destination image is used to continue to do the subsequent convolutions (Figure 4). After each convolution, we use a pooling function to reduce the spatial size of the convolved feature. The pooling method we used is called Max Pooling, which returns the maximum value from the portion of the image covered by the kernel. This pooling method includes only the maximum pixel into the destination image instead of completing the matrix multiplication again. Between each convolution, a Batch Normalization function is added to normalize the data from the previous convolution ensuring normalization of the entire training process (see Table 2 for environment specifications).

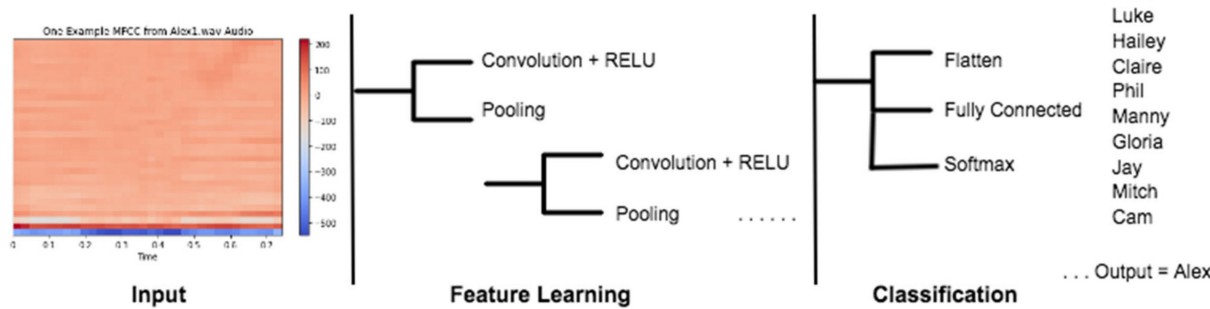

**Figure 3.** The process to train and classify the voices from Modern Family (Adapted from https://towardsdatascience.com/a-comprehensive-guide-to-convolutional-neural-networks-the-eli5-way-3bd2b1164a53, accessed on 5 June 2020).

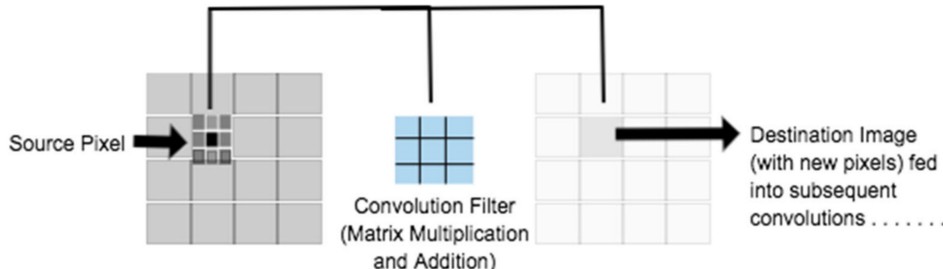

**Figure 4.** Feature learning and convolution (Adapted from https://towardsdatascience.com/applied-deep-learning-part-4-convolutional-neural-networks-584bc134c1e2, accessed on 5 June 2020).

**Table 2.** Environmental specifications for the Basic and Advanced CNN models.

| Models | Basic CNN Model | | | Advanced CNN Model | | |
|---|---|---|---|---|---|---|
| Batch size | 32 | | | 32 | | |
| Iteration | 1262 | | | 1262 | | |
| Duration of each Iteration | 8–9 ms | | | 16–17 ms | | |
| Epochs | 50 | | | 50 | | |
| Duration of each epoch | 10–12 s | | | 20–21 s | | |
| Learning Rate | 0.01 | | | 0.01 | | |
| Optimizer function | Adam | | | Adam | | |
| Loss function | Sparse categorical cross entropy | | | Sparse categorical cross entropy | | |
| Hidden layer details | Hidden layer | Hidden units | Activation function | Hidden layer | Hidden units | Activation function |
| | | | | Convolutional layer | 32 | Rectified Linear Unit |
| | | | | Convolutional layer | 32 | Rectified Linear Unit |
| | | | | Convolutional layer | 64 | Rectified Linear Unit |
| | Convolutional layer | 32 | Rectified Linear Unit | Convolutional layer | 64 | Rectified Linear Unit |
| | | | | Convolutional layer | 128 | Rectified Linear Unit |
| | | | | Convolutional layer | 128 | Rectified Linear Unit |
| | Dense layer | 1024 | Rectified Linear Unit | Dense layer | 1024 | Rectified Linear Unit |

## 3. Result

### 3.1. Convolutional Neural Network Model: Testing

In the classification stage (refer to Figure 3), we followed the process of: Flatten, Fully Connected Layers, and Softmax. The "Flatten" process reduces the three-dimensional matrix into a one-dimensional vector. The "Fully Connected Layers" process ensures that all inputs from one layer are all connected to the activation units of the next layer [39].

Finally, the Softmax process is implemented, which extracts the predicted class of each of the ten voices.

For the basic CNN model we started with a simple model that had one convolutional layer with 32 filters, (3, 3) kernel size, (1, 1) strides, ReLU activation, and "same" padding (see Appendix A). Flowing by a pooling layer with (2, 2) pool size. Then after the flatten layer, we had two general Neural Network layers, one had 1024 neurons with ReLU activation, and the output layer has 10 neurons with softmax activation. We used "adam" as the optimizer and "sparse_categorical_crossentropy" as the loss function [40]. Figure 5A is a plot of the loss (i.e., the gap between real data and predicted data) of training data (in blue) and the loss of validation data (in orange). Figure 5B is the chart for the accuracy of the training data (in blue) and the accuracy of the validation data (in orange). After 50 epochs training of the basic CNN model, the testing accuracy was 89.02% and was somewhat volatile (i.e., random shifts in performance).

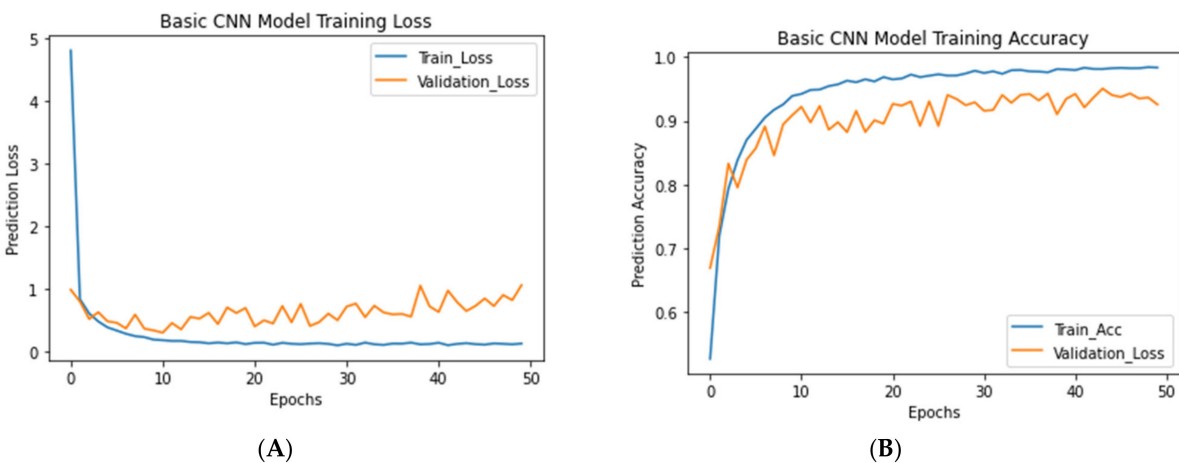

**Figure 5.** Basic CNN Model Testing. (**A**). Prediction losses as a function of epochs. (**B**). Prediction accuracy as a function of epochs.

With the advanced CNN model, the model structure corresponding to the advanced CNN model can be found in Figure 6 (see source code and parameters table in the Appendix A). We started an advanced model with one convolutional layer with 32 filters, (3, 3) kernel size, (1, 1) strides, ReLU activation, and "same" padding, and added one batch normalization. The second convolution layer also had 32 filters, (3, 3) kernel size, (1, 1) strides, ReLU activation, and "same" padding and one batch normalization. Then, the network flowed by a pooling layer with (2, 2) pool size. After that, the third and fourth convolutional layer with 64 filters, (3, 3) kernel size, (1, 1) strides, ReLU activation, and "same" padding, added one batch normalization as well. After the second max-pooling layer, the last two convolutional layers had 28 filters, (3, 3) kernel size, (1, 1) strides, ReLU activation, and "same" padding, and added one batch normalization and one max pooling layer. We also used "adam" as the optimizer and "sparse categorical crossentropy" as the loss function [40]. After the "flatten" layer, we had two general Neural Network layers, one has 1024 neurons with ReLU activation, and the output layer has 10 neurons with Softmax activation [40].

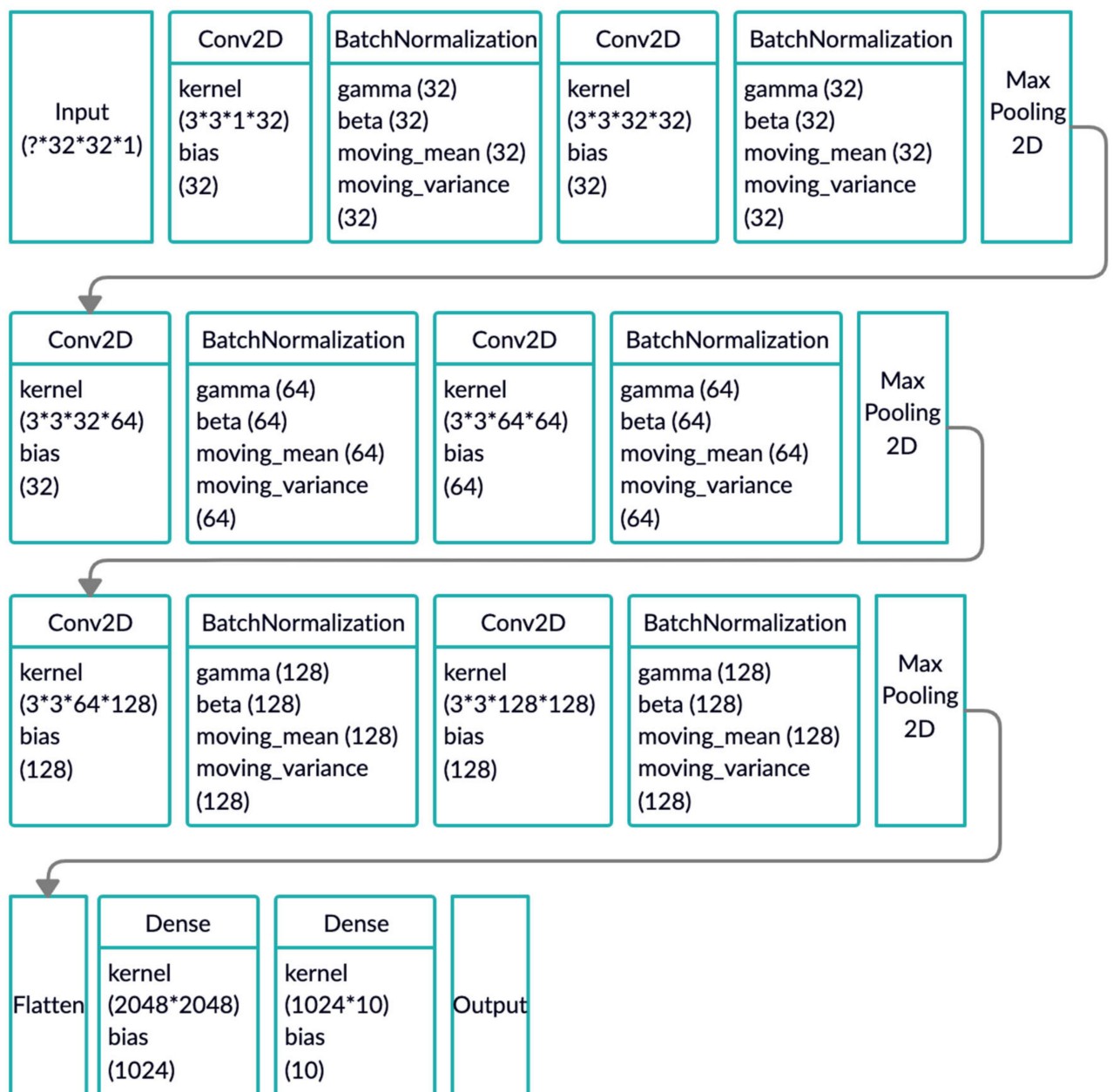

**Figure 6.** Model structure corresponding to the advanced CNN model implemented in the current study (see code in the Appendix A).

We can see that using the advanced model, after 50 epochs of training, the testing accuracy of the advanced CNN model is 99.86 and rarely volatile (i.e., random shifts in performance) (Figure 7A,B; Table 3). Figure 7C shows the confusion matrix associated with the testing data. Numbers in the figure correspond to frequency counts. Numbers on the diagonal are correct classifications and numbers off the diagonal are incorrect classifications. For example, it can be seen that Alex was incorrectly classified twice: once as Haley and once as Manny.

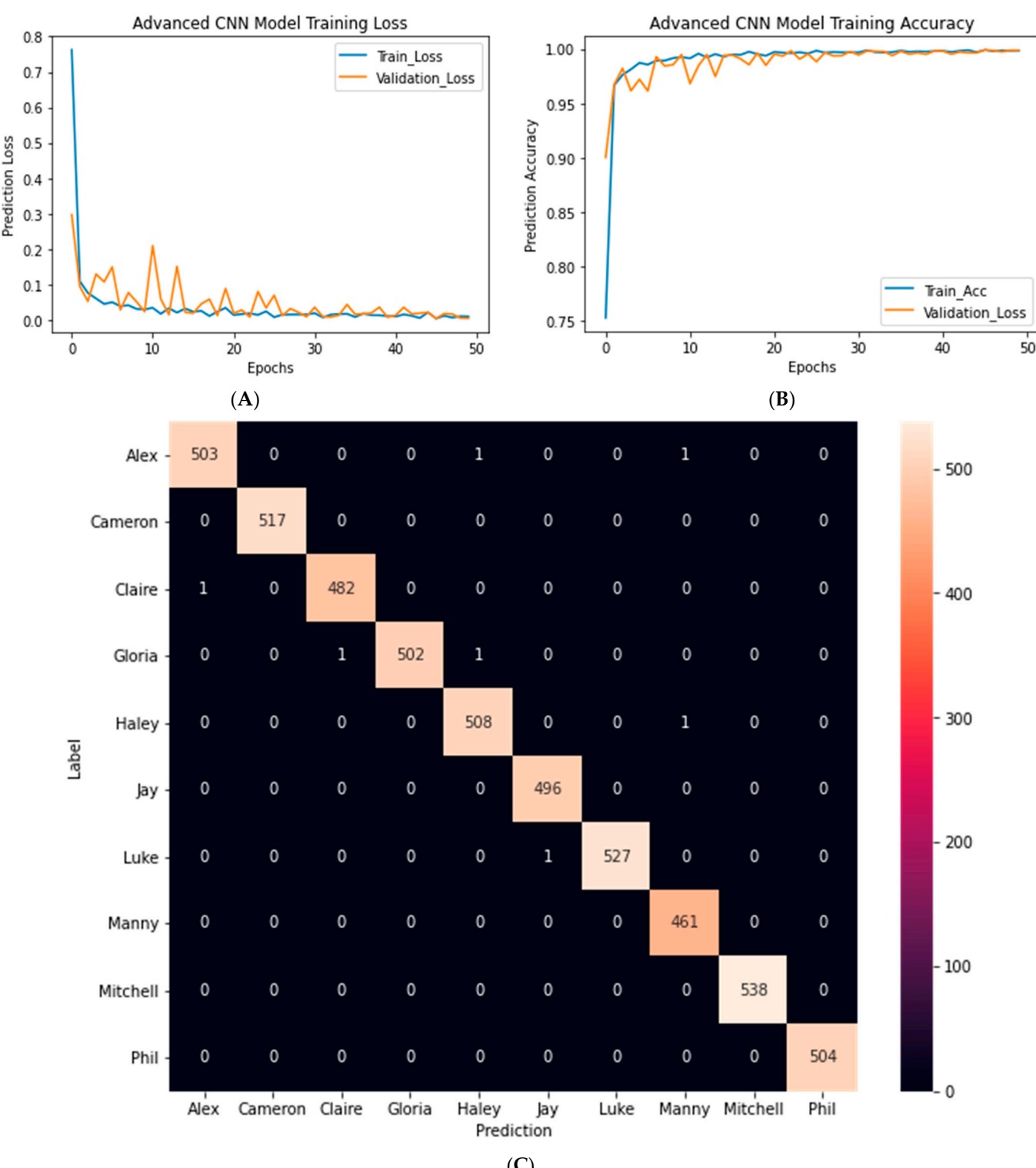

**Figure 7.** Advanced CNN Model Testing. (**A**). Prediction losses as a function of epochs. (**B**). Prediction accuracy as a function of epochs. (**C**). Confusion matrix of the Advanced CNN model in the test set (10% of the data). Numbers that fall on the diagonal are correctly classified voices. Numbers that fall off the diagonal are incorrectly classified voices. Label = Actual Voice. Prediction = Predicted Voice.

**Table 3.** The accuracy of each class in the test set in the Advanced CNN model.

| Class | Accuracy (%) |
|---|---|
| Alex | 99.60 |
| Cameron | 100 |
| Claire | 99.79 |
| Gloria | 99.60 |
| Haley | 99.80 |
| Jay | 100 |
| Luke | 99.81 |
| Manny | 100 |
| Mitchell | 100 |
| Phil | 100 |

### 3.2. Model Overfitting Considerations

Lastly, we evaluated the CNN model to test if it was overfitting. According to Ying [41], overfitting is a common problem in supervised machine learning, and is present when a model fits well to observed data (i.e., training data) but not test data. In general, overfitting of a model means that our model does not generalize well from training data to unseen data [42]. There are some known causes of this phenomenon [32,41]. For example, noise in the original dataset may be learned through the training process but then cannot be captured/modelled in the testing phase. Secondly, there is always a trade-off in the complexity of machine learning (or any predictive algorithm for that matter), whereby bias and variance are forms of prediction error in the learning process. When an algorithm has too many inputs, the consistency of accurate output tends to be less stable [41,42]. Here we took several steps to decrease overfitting. To minimize inherent noise, we implemented a data cleaning process to remove some potential noise in our original dataset. This produced a "cleaned" audio signal that served in the training and testing phases. Secondly, we created a larger dataset in the training phase (i.e., ×50), which serves to provide more training for our model. Thirdly, we started with a very simple model (look at the basic model and advanced model comparison) to serve as a benchmark. Finally, we evaluated the extent to which the model performed much better on the training set than on the test set. This was not an issue as our test set resulted in an accuracy of 99%.

Generalization testing: After using our data to train and test, we then tested the extent to which our model could be generalized to other datasets. We chose the VoxCeleb dataset (also 10 people with 30 audios for each), which has hundreds of voice recordings of celebrities. We wanted to use those voice audios to test if our machine learning model performs overfitting or only works for our own data. Our process was as follows: We downloaded the audios from here: https://www.robots.ox.ac.uk/~vgg/data/voxceleb/= (accessed on 24 June 2020) and we extracted 30 audios from each random person (10 in total) for 300 files total. After using our advanced machine learning model to train the data, we demonstrated that when using the advanced model, after 50 epochs of training, the testing accuracy of the CNN model is 99.95% (Figure 8A,B).

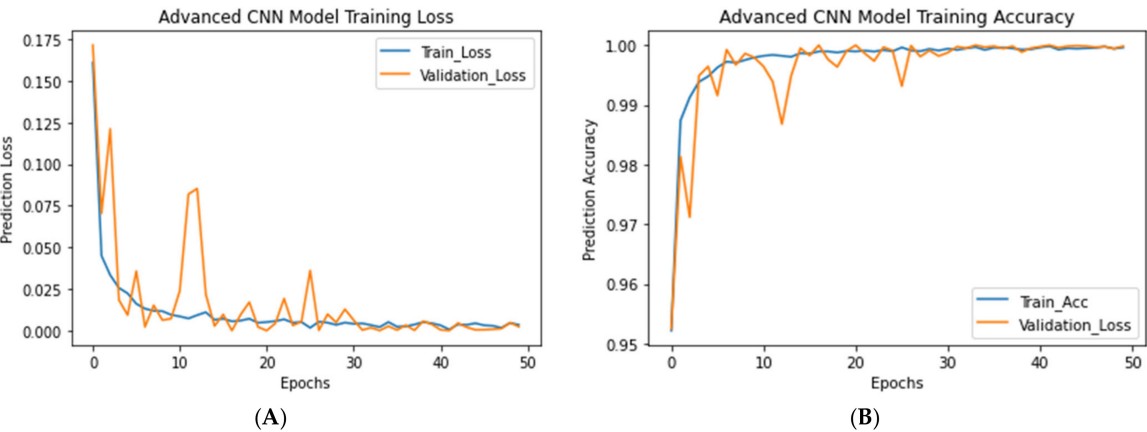

(**A**)                              (**B**)

**Figure 8.** Advanced CNN model applied to new data. (**A**). Prediction losses as a function of epochs. (**B**). Prediction accuracy as a function of epochs.

In summary, even with the use of online audios (wav. format), our audio clean function and machine learning model continued to work well, which indicates that they are not only specific to the audios used in this study, but also for general audios unfamiliar to the model.

## 4. Discussion

Here we provide a proof of concept for the inclusion of "voice familiarity" as a potential hearing aid feature. Using a machine learning approach, we demonstrate the feasibility of a convolution neural network (CNN) model to detect and recognize auditory input via conversion to a spectrogram. We also demonstrate that the implemented model, once trained, can accurately classify a family of speakers (i.e., the 10 members of the Modern Family television series) to an accuracy level of >99%. We discuss the implications of the work in the context of hearing aid feature development.

In this work, we were concerned with addressing two main goals. Firstly, we set out to determine the feasibility of creating a machine learning classifier to detect and recognize auditory input. After much exploration into the types of neural network models that can be utilized to implement the desired work (i.e., classification of voices), we settled on the CNN model. The rationale for this was the image feature component of the CNN that aligned well with the image feature of audio files, namely the spectrogram. We know that an audio signal, and by transformation, the corresponding spectrogram, carries a staggering amount of information to the listener [43] for a CNN application of emotion recognition from a spectrogram. While alternative neural networks exist (e.g., recurrent neural networks, RNN, which are particularly useful in the prediction of time-series information) [44], the feedforward nature of CNN, in conjunction with the multiple convolutions in the advanced model made this the ideal starting point. Furthermore, while RNN (and hybrid models) can be suitable applications for audio classification and verification, these approaches require substantially more computation power and much larger datasets. According to Zhang et al. [29] the hybrid models can be particularly problematic as different sets of parameters/hyperparameters may be selected at each stage of the processes, making the interpretation difficult and possibly not optimal for the final solution. Given our motivation to (eventually) implement the voice familiarity classification to a hearing device, a CNN is an ideal approach as we (1) have maximum control/interpretation over the feature implementations at each step, (2) can train, validate and test the CNN with a relatively small dataset, and (3) do not require egregiously large computational power. An additional advantage of using the "image-based" spectrogram as the input to the CNN model, was the numerous possible features that can be explored, modified, extracted, characterized, etc., in future work. So, for example, if one was interested in determining which feature of the spectrogram "most" contributes to voice familiarity, a systematic approach to the modification of each of the components/features of the spectrogram can be taken. There is currently limited information on the various components of the spectrogram that contribute to voice recognition/familiarity, and the model tested here provides an avenue for that work to be explored even further.

Secondly, we aimed to test the accuracy with which the CNN model could be trained to classify a series of voices, in this case, the voices from the popular television series Modern Family. In this regard, we considered three pieces of evidence. Firstly, we established that a basic CNN model (i.e., one with a single convolution layer and no normalization) could classify the voices, and indeed the basic CNN model achieved an accuracy of 89%. Secondly, we created and tested a more advanced CNN model (i.e., one with six convolution layers and normalization at each layer that fed forward as the input to the next convolution), which resulted in a "deeper" neural network, and subsequently greater classification accuracy (i.e., 99%) as compared to the basic CNN model (i.e., 89%). Thirdly, we determined that the advanced CNN model was not overfitting our data, via examination of the training, validation and testing inputs/outputs, but indeed was a reasonable neural network solution to voice classification. While our findings in voice classification are not novel within the machine learning domain, they add to the relatively dearth speech recognition literature [30]

and are a necessary first step (i.e., replication and implementation of machine learning toward voice classification) towards building novel voice classification to extract familiar voice features and apply them to novel voices in real time. In line with Nassif's et al [30], recommendation, future work should begin to explore recurrent neural networks (RNN) and hybrid approaches in the speech recognition domain (although see Zhang et al. [29] and You et al. [32] for challenges with these approaches). Indeed, Al-Kaltakchi, Abdullah, Woo and Dlay [45] recently used a hybrid approach to evaluate machine learning capacity in speaker identification in a variety of environments (i.e., background noise) and while there is some modest variability with respect to the various factors the authors explored, the speaker identification accuracies range from 85–97% (see Figure 2). Similar to the current work (i.e., Generalization testing), Yadav and Rai [46] examined speaker identification using a CNN with the VoxCeleb database. With the additional implementation of Softmax and centrer loss features, these authors reported speaker identification accuracy around 90%, and thus our findings are within the range of previous work. The extent to which these hybrid models can perform under various real-time constraints remains to be seen. In summary, these results provide confidence in the proof of the concept that machine learning approaches can be used to classify voices of varying ages, genders, accents, etc.

The CNN model described here, and the potential for subsequent development and modification of the neural network provides some exciting new avenues of research to explore. For example, with emerging developments in the text-to-speech machine learning domain [47], we may be able to develop a greater understanding of how humans "lock on" to speech, distinguish between familiar and unfamiliar voices and/or the robustness of our brains to adapt to changes in familiar voices via aging or sickness (see Mohammed et al. [48] for an application of machine learning to classification of voice pathology), just to name a few. Of particular interest to our group is the extent to which an unfamiliar voice can be modified to resemble a familiar voice, and subsequently decrease listening effort for the hearing-impaired individual.

### 4.1. Limitations and Future Directions

Obviously, there is much to be discovered before such an endeavor is, in practice, feasible. For example, it remains to be seen if hearing aids will have the computing power of modern PC machines to store features of a familiar voice. Even if such storage is possible, it remains likely a few years hence before real-time processing and voice conversion can be realized. Furthermore, there is the nagging unknown of whether people will be able to tolerate or accept a voice from a stranger that sounds similar and familiar to your partner or spouse, even if it does lead to easier listening and better performance. It is conceivable that this might be very strange for the listener. Nevertheless, the work outlined here was a necessary first step along this path.

### 4.2. Bigger Picture: What Does Monday Look Like?

So, back to considering the individual with hearing loss sitting in the restaurant with the friendly patron at the table next to them. With all other current hearing aid features exhausted, including noise reduction, directional microphones, etc., the individual with hearing loss may find that they are continuing to struggle to have a conversation. Now imagine that the individual could adjust their hearing aid to an alternate setting, that includes all of the aforementioned hearing aid features in addition to a voice feature "maximizer" that applied a set of familiar voice features to the incoming signal. In this case, the incoming "unfamiliar" voice can capitalize on "familiar" features, resulting in a reduced cognitive load and listening effort.

### 5. Conclusions

Here we provide a proof of concept with respect to including "voice familiarity" as a feature in a hearing aid. We provide evidence for the feasibility of a machine learning classifier to identify speakers with high accuracy. The next steps in this line of research



include voice conversion, real-time functioning, software advancements to accommodate such technology, and ultimately, evidence for the behavioral benefits associated with such hearing aid technology.

**Author Contributions:** Conceptualization, W.H. and J.C.; methodology, Q.S. and X.X.; software, Q.S. and X.X.; validation, Q.S. and X.X.; resources, W.H. and J.C.; writing—original draft preparation, W.H. and J.C.; writing—review and editing, W.H., J.C., Q.S. and X.X.; visualization, Q.S. and X.X.; supervision, W.H. and J.C. All authors have read and agreed to the published version of the manuscript.

**Funding:** This research was partially funded by the Natural Sciences and Engineering Council of Canada, Canada Research Chair grant to author J.C.

**Institutional Review Board Statement:** Not applicable.

**Informed Consent Statement:** Not applicable.

**Data Availability Statement:** All links to publicly archived datasets are provided in the text.

**Conflicts of Interest:** The authors declare no conflict of interest.

**Appendix A**

```python
import numpy as np
import pandas as pd
import scipy.io as sio
import matplotlib.pyplot as plt
import tensorflow as tf
import librosa
from sklearn.utils.class_weight import compute_class_weight
from tensorflow.keras.layers import Input, Conv2D, Dense, Flatten, Dropout
from tensorflow.keras.layers import GlobalMaxPooling2D, MaxPooling2D, BatchNormalization
from tensorflow.keras.models import Model, save_model, load_model
from tensorflow.keras.utils import to_categorical
from tqdm import tqdm
```

**Figure A1.** Dependencies and Libraries used in this experiment.

```python
def envelope(y, rate, minThreshold, maxThreshold):
    mask = []
    y = pd.Series(y).apply(np.abs)
    y_mean = y.rolling(window=int(rate/10), min_periods=1, center=True).mean()
    for mean in y_mean:
        if mean > minThreshold and mean < maxThreshold:
            mask.append(True)
        else:
            mask.append(False)
    return mask
```

**Figure A2.** Envelope Function Code.

```python
def reshape(X, y):
    X, y = np.array(X), np.array(y)
    X = X.reshape(X.shape[0], X.shape[1], X.shape[2], 1)
    return X, y

def sampling(data, rate):
    step = int(rate/2)
    index = np.random.randint(0, data.shape[0]-step)
    sample = data[index:index + step]
    return sample

def pickRandomFile():
    random_class = np.random.choice(class_dist.index, p=[0.1]*10)
    filename = np.random.choice(df[df.label == random_class].index)
    return filename, random_class

def extract_feature():
    X = []
    y = []
    n_samples = int(df['length'].sum()) * 50
    for _ in tqdm(range(n_samples)):
        filename, random_class = pickRandomFile()
        rate, data = sio.wavfile.read(DIR+'MF10PeopleClean/'+filename)
        sample = sampling(data, rate)
        S = librosa.feature.melspectrogram(y=sample, sr=rate, hop_length=700)
        x = librosa.feature.mfcc(n_mfcc=32, n_fft=512, S=librosa.power_to_db(S))
        X.append(x)
        y.append(classes.index(random_class))
    X, y = reshape(X, y)
    return X, y
```

**Figure A3.** Feature Extraction Code.

```python
def BasicCNNModel(X):
    i = Input(shape=X[0].shape)
    x = Conv2D(32, (3, 3), activation='relu', padding='same')(i)
    x = MaxPooling2D((2, 2))(x)
    x = Flatten()(x)
    x = Dense(1024, activation='relu')(x)
    x = Dense(10, activation='softmax')(x)
    model = Model(i, x)
    return model
```

**Figure A4.** Basic CNN Model Code.

```python
def AdvancedCCNModel(X):
    i = Input(shape=X[0].shape)
    x = Conv2D(32, (3, 3), activation='relu', padding='same')(i)
    x = BatchNormalization()(x)
    x = Conv2D(32, (3, 3), activation='relu', padding='same')(x)
    x = BatchNormalization()(x)
    x = MaxPooling2D((2, 2))(x)
    x = Conv2D(64, (3, 3), activation='relu', padding='same')(x)
    x = BatchNormalization()(x)
    x = Conv2D(64, (3, 3), activation='relu', padding='same')(x)
    x = BatchNormalization()(x)
    x = MaxPooling2D((2, 2))(x)
    x = Conv2D(128, (3, 3), activation='relu', padding='same')(x)
    x = BatchNormalization()(x)
    x = Conv2D(128, (3, 3), activation='relu', padding='same')(x)
    x = BatchNormalization()(x)
    x = MaxPooling2D((2, 2))(x)
    x = Flatten()(x)
    x = Dense(1024, activation='relu')(x)
    x = Dense(10, activation='softmax')(x)
    model = Model(i, x)
    return model
```

**Figure A5.** Advanced CNN Model Code.

| Layer (type) | Output Shape | Param # |
|---|---|---|
| input_2 (InputLayer) | [(None, 32, 32, 1)] | 0 |
| conv2d_1 (Conv2D) | (None, 32, 32, 32) | 320 |
| batch_normalization (BatchNo | (None, 32, 32, 32) | 128 |
| conv2d_2 (Conv2D) | (None, 32, 32, 32) | 9248 |
| batch_normalization_1 (Batch | (None, 32, 32, 32) | 128 |
| max_pooling2d_1 (MaxPooling2 | (None, 16, 16, 32) | 0 |
| conv2d_3 (Conv2D) | (None, 16, 16, 64) | 18496 |
| batch_normalization_2 (Batch | (None, 16, 16, 64) | 256 |
| conv2d_4 (Conv2D) | (None, 16, 16, 64) | 36928 |
| batch_normalization_3 (Batch | (None, 16, 16, 64) | 256 |
| max_pooling2d_2 (MaxPooling2 | (None, 8, 8, 64) | 0 |
| conv2d_5 (Conv2D) | (None, 8, 8, 128) | 73856 |
| batch_normalization_4 (Batch | (None, 8, 8, 128) | 512 |
| conv2d_6 (Conv2D) | (None, 8, 8, 128) | 147584 |
| batch_normalization_5 (Batch | (None, 8, 8, 128) | 512 |
| max_pooling2d_3 (MaxPooling2 | (None, 4, 4, 128) | 0 |
| flatten_1 (Flatten) | (None, 2048) | 0 |
| dense_2 (Dense) | (None, 1024) | 2098176 |
| dense_3 (Dense) | (None, 10) | 10250 |

```
Total params: 2,396,650
Trainable params: 2,395,754
Non-trainable params: 896
```

**Figure A6.** Advanced Model CNN Structure Parameters Table.

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
