# Peer review of "On a Vector towards a Novel Hearing Aid Feature: What Can We Learn from Modern Family, Voice Classification and Deep Learning Algorithms"

_applsci, doi:10.3390/app11125659_

Round 1

Reviewer 1 Report

The subject adressed in this subject is interesting and relevant for the voice processing field. 

After reading and analyzing the paper, I can make the following observations:

- The Discussion includes the presentation of this study’s aims and results. Authors need to performe comparison of their results with more similar studies.
- To test the accuracy of the classification it would be useful to check the operation of the classifier in case of voices different from those used in the training process. In these cases the classifier should not identify any speaker (output 0.0, ...., 0).
- The paragraph between lines 68-75 needs to be revised in order to enhance its clarity.
- In the Feature Extraction section it is not very clear how the trainig samples are generated. The authors state that the length of each training sample is half of the sample rate that means an equivalent of 0.5 second recording. This length would lead to 2018 trainig samples for the 1009 seconds total audio recordings.

In conclusion, I recommend to reconsider the article after major revision according to the previous observations.

Reviewer 2 Report

In the paper, the authors provide a proof of concept with respect to including ‘voice familiarity’ as a feature in a hearing aid. The authors provide evidence for the feasibility of a machine learning classifier to identify speakers with high accuracy.

The paper looks like a technical report about using CNN and machine learning methods for voice classification. 

The paper lacks scientific novelty. The motivation of the paper is not clear. The results of the paper are not clear and should be compared with related approaches. The hearing aid relevance of the proposed approach should be better commented on.  The authors write about classification but they don't present a description of classes and resulting accuracy per class. A lot of technical details are missed.

The authors should present their contribution to the model proposed in Figure 6 clearly.

The authors should present a detailed description of the data used for train and testing. 

The organization and presentation of the paper should be improved. 

Reviewer 3 Report

The article demonstrates the feasibility of a convolution neural network (CNN) model to detect and recognize auditory input via conversion to a spectrogram. Also, the implemented model, once trained, can accurately classify a family of speakers (i.e., the 10 members of the Modern Family television series) to an accuracy level of > 99%. The authors investigate the perspective of voice familiarity as a possible hearing aid feature with the help of CNN. 
The paper title is appropriate and, the abstract contains the necessary information. 
In the introduction, the authors use a sufficient number of references to give the necessary background around the topic of hearing aid.  However, some of them are not recent and should be replaced. The main contribution is mentioned at the end of the Introduction.  The methodology is clear. The Pipeline for preprocessing and the voice classification model are illustrated in the first figure.  
The authors justify the choice of the current data set. The rationale for choosing these voices is that they exhibit diversity in gender, age, accent, are easily accessible, and provide future opportunities to explore developmental factors (i.e., there are 10 years/seasons worth of voices accessible across the age range). The experiment setup description for each component is sufficient and the necessary codes for reproducibility are given.
The research outcomes are clearly presented and discussed in relation to other research studies. The authors depict the training and validation loss, and accuracy performance. 
The technical contribution is limited. The CNN model, both a simple and advanced version of it, is utilized for the desired feature extraction. Nonetheless, the authors need to elaborate, why the CNN model was selected. Also, which other deep models could be used to achieve the same goal. 
 Kindly, mention the environment's specifications used in the experiments and summarize in a table the hyperparameters tuned for the CNN model used in this study.                                                           For discussion, the limitations and potential issues of this study should be recorded.                                                                                                  The Conclusions need considerable enhancement and connection with the contribution and the results of this work. Some future directions are simply stated. I recommend you elaborate further on them.
The authors should carefully proofread the text for grammatical errors correction.

As a final note, the title of the manuscript is "Voice Classification and Deep Learning Algorithms". It is different from the one stated in the submission form, namely, "On a vector towards a novel hearing aid feature: What can we learn from Modern Family, voice classification and deep learning algorithms".  Could you justify why this mismatch occurred?

Round 2

Reviewer 1 Report

Some of the previously formulated requirements have been resolved, but there are still some observations that should be taken into account:

- In the Introduction section (lines 84 to 90) what the "implicit training" and "explicit training" refer to? However this paragraph needs some reformulations because it is difficult to be understood.

-  It is still unclear why the number of training samples is 50450. If the total length of audio data was 1009 seconds and each training sample was chosed to have 0.5 second length, the result would be a number of 2018 training samples. Multiplied by 50 epochs would obtain 100900 samples not 50450.

- In the confusion matrix presented in Figure 7C, what means the numbers? Are they the numbers of correctly/incorrectly classified samples? If yes, what data was used for confusion matrix generation: training, validation, test  or all data? It would be useful to describe in detail how this matrix was generated.

In conclusion, I recommend to revise the article according to previous observations, in order to be accepted for publish

Author Response

- In the Introduction section (lines 84 to 90) what the "implicit training" and "explicit training" refer to? However this paragraph needs some reformulations because it is difficult to be understood.

Response: We have re-worked this paragraph to make the literature on voice familiarity more clear to the reader. We have removed the mention of implicit vs. explicit training as these methods are not critical to the genesis of the paper.

-  It is still unclear why the number of training samples is 50450. If the total length of audio data was 1009 seconds and each training sample was chosed to have 0.5 second length, the result would be a number of 2018 training samples. Multiplied by 50 epochs would obtain 100900 samples not 50450.

Response: We have a total number of 50450 samples (i.e., 1009 seconds x 50 epochs). The 0.5 seconds mentioned in the previous version was with respect to the sampling rate of the waveform that was fast fourier transformed into a spectrogram. We have removed this latter information as it is not necessary for replication of the current work and is standard practice (i.e., when converting from time domain to frequency domain max bandwidth = sampling rate * 0.5). 

- In the confusion matrix presented in Figure 7C, what means the numbers? Are they the numbers of correctly/incorrectly classified samples? If yes, what data was used for confusion matrix generation: training, validation, test  or all data? It would be useful to describe in detail how this matrix was generated.

Response. We have updated the figure legend to provide more details about the confusion matrix. Specifically, the numbers correspond to frequency counts of classification of a voice via the CNN. Numbers on the diagonal are correct. Numbers off the diagonal are incorrect classifications. The confusion matrix is based on the testing data only.

Figure 7C shows the confusion matrix associated with the testing data. Numbers in the figure correspond to frequency counts. Numbers on the diagonal are correct classifications and numbers off the diagonal are incorrect classifications. For example, it can be seen that Alex was incorrectly classified two times: once as Haley and once as Manny.”

Reviewer 2 Report

The idea stated by the authors is not novel. The idea of saving the auditory profile has been seen in many products in the past. For example:

https://petralex.pro/en

As for ML-based techniques used by the authors, it's not novel and well described in the papers. For example:

Shamir, L.; Yerby, C.; Simpson, R.; von Benda-Beckmann, A.M.; Tyack, P.; Samarra, F.; Miller, P.; Wallin, J. Classification of large acoustic datasets using machine learning and crowdsourcing: Application to whale calls. Acoust. Soc. Am. 2014, 135, 953–962.

Thus, the paper doesn't have any scientific or technical novelty.

Author Response

The idea stated by the authors is not novel. The idea of saving the auditory profile has been seen in many products in the past. For example:

https://petralex.pro/en

As for ML-based techniques used by the authors, it's not novel and well described in the papers. For example:

Shamir, L.; Yerby, C.; Simpson, R.; von Benda-Beckmann, A.M.; Tyack, P.; Samarra, F.; Miller, P.; Wallin, J. Classification of large acoustic datasets using machine learning and crowdsourcing: Application to whale calls. Acoust. Soc. Am. 2014, 135, 953–962.

Thus, the paper doesn't have any scientific or technical novelty.

Response:

We agree with the reviewer, and have tried to be clear, that our methods (scientific and technical) are indeed replications of previous work. While not ‘scientifically or technically’ novel, we have contextualized our work in two ways: 1) the need for additional replications/demonstrations of voice classification approaches (as recommended in Nassif et al., 2019) and 2) discussions about potential next step applications of voice classification technology (namely, voice familiarity as a hearing aid feature).  We have updated the manuscript to be clear in our objective of this work.

Abstract: (1) Background: The application of machine learning techniques in the speech recognition literature has become a large field of study. Here, we aim to 1) expand the available evidence for the use of machine learning techniques for voice classification and 2) discuss the implications of such approaches towards the development of novel hearing aid features (i.e., voice familiarity detection). To do this, we built and tested a Convolutional Neural Network (CNN) Model for the identification and classification of a series of voices, namely the 10 cast members of the popular television show “Modern Family”.  (2) Methods: Representative voice samples were selected from Season 1 of Modern Family (N = 300; 30 samples for each of the classes of the classification in this model, namely Phil, Claire, Hailey, Alex, Luke, Gloria, Jay, Manny, Mitch, Cam). The audio samples were then cleaned and normalized. Feature extraction was then implemented and used as the input to train a basic CNN model and an advanced CNN model. (3) Results: Accuracy of voice classification for the basic model was 89%. Accuracy of the voice classification for the advanced model was 99%.; (4) Conclusions: Greater familiarity with a voice is known to be beneficial for speech recognition. If a hearing aid can eventually be programmed to recognize voices that are familiar or not, perhaps, too, it can apply familiar voice features to improve hearing performance. Here we discuss how such machine learning applied to voice recognition is a potential technological solution in the coming years.

Summary. Here we implement a machine learning approach to speaker identification to expand the available evidence in this space. In addition, we discuss the implications of such work in the context of eventually including ‘voice familiarity’ as a feature in a hearing aid.  In doing so, we provide additional support regarding the feasibility and consistency of a machine learning classifier to detect and recognize auditory input, in this case, the voices from the popular television series Modern Family. Furthermore, we provide a novel line of inquiry to be pursued by researchers in the hearing aid development space, namely capitalizing on the benefits of voice familiarity.

Round 3

Reviewer 1 Report

I read the revised version of the article and I noticed that all of my observations were properly addressed.
In conclusion, I recommend to accept the article for publication.